# SEED-STORY: MULTIMODAL LONG STORY GENERATION WITH LARGE LANGUAGE MODEL

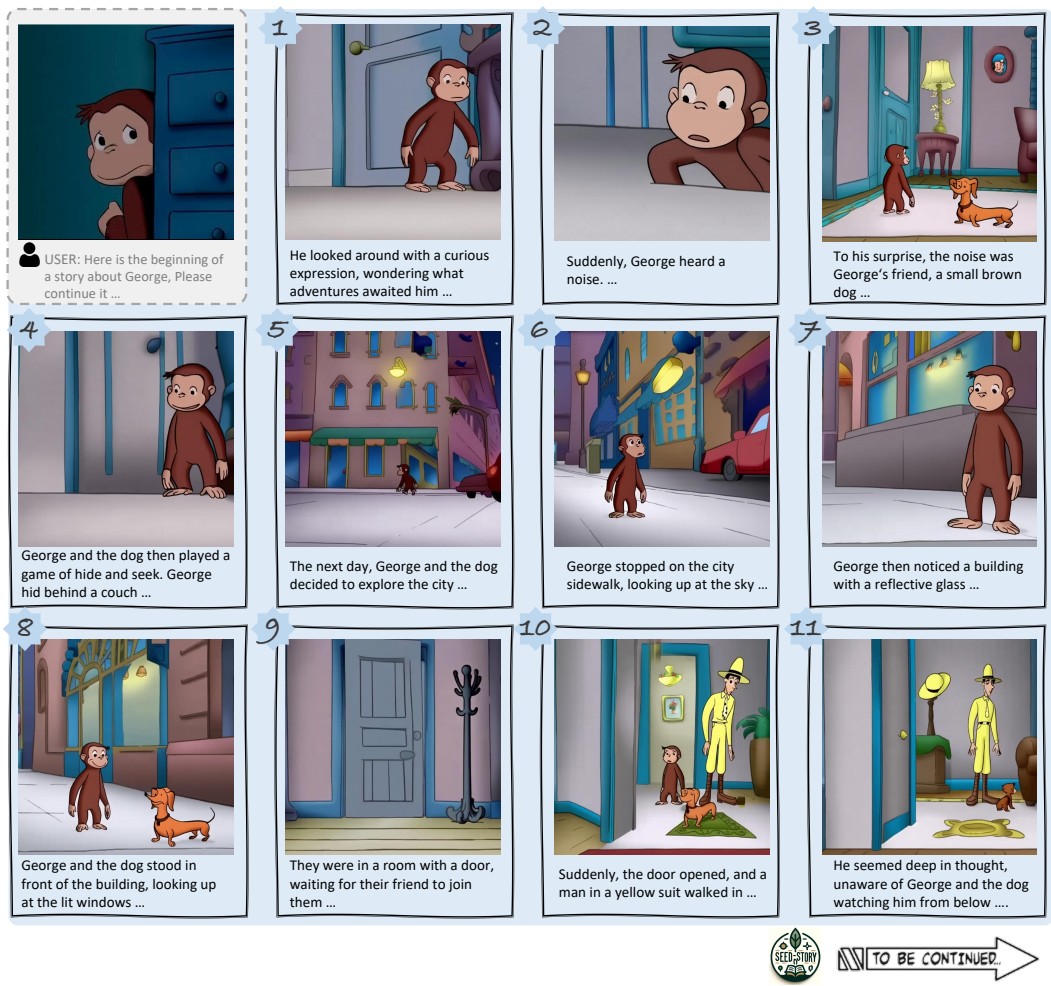

Figure 1: The introduced SEED-Story, powered by MLLM, is capable of generating **multimodal long stories** based on user-provided images and texts as the story's beginning. The generated story features rich and coherent narrative texts, accompanied by images that maintain consistency in characters and style. The story can span up to 25 multimodal sequences (see Appendix), even though we only use a maximum of 10 sequences for training.

## ABSTRACT

With the remarkable advancements in image generation and open-form text generation, the creation of interleaved image-text content has become an increasingly intriguing field. Multimodal story generation, characterized by producing narrative texts and vivid images in an interleaved manner, has emerged as a valuable and practical task with broad applications. However, this task poses significant challenges, as it necessitates the comprehension of the complex interplay be-

tween texts and images, and the ability to generate long sequences of coherent, contextually relevant texts and visuals. In this work, we propose SEED-Story, a novel method that leverages a Multimodal Large Language Model (MLLM) to generate extended multimodal stories. Our model, built upon the powerful comprehension capability of MLLM, predicts text tokens as well as visual tokens, which are subsequently processed with an adapted visual de-tokenizer to produce images with consistent characters and styles. We further propose multimodal attention sink mechanism to enable the generation of stories with up to 25 sequences (only 10 for training) in a highly efficient autoregressive manner. Additionally, we present a large-scale and high-resolution dataset named StoryStream for training our model and quantitatively evaluating the task of multimodal story generation in various aspects. All models, training and inference codes are released in `https://anonymous.4open.science/r/SEED-Story/`.

## 1 INTRODUCTION

Interleaved image-text data is ubiquitous on the internet, characterized by multiple images interspersed with text. In recent years, there has been a surge of interest in generating interleaved image-text content Tian et al. (2024); Ge et al. (2024); Aiello et al. (2023); Dong et al. (2023); Team (2024), driven by the remarkable advancements in image generation Rombach et al. (2022); Lin et al. (2023); Chen et al. (2023); Patashnik et al. (2021); Wang et al. (2023) and open-form text generation Touvron et al. (2023); Taori et al. (2023); Zheng et al. (2023). This has given rise to **Multimodal Story Generation**, an intriguing and valuable task that involves the generation of a sequence of consecutive storylines along with their corresponding vivid images in an interleaved manner, similar to that of a serialized comic.

Different from **Personalized Story Visualization** Ruiz et al. (2023); Avrahami et al. (2024); Tewel et al. (2024), which aims to generate consistent images based on the provided captions following the pattern of text-to-image generation, multimodal story generation poses a more significant challenge due to the complexity of the inputs and the high demands of the outputs. Firstly, this task necessitates a thorough comprehension of interleaved data, where text is not only abstract and narrative in nature, but also deeply intertwined with complex images. The model must be adept at deciphering the intricate relationships between images and texts to maintain a coherent narrative flow. Secondly, this task requires the generation of not only a plausible text plot, but also visually captivating images that are consistent in characters and styles. The model should be capable of achieving coherence in the generation of both text and visuals, ensuring an engaging storytelling output.

Recently, Multimodal Large Language Models (MLLMs) Li et al. (2023); Zhu et al. (2023a); Peng et al. (2023a); Bai et al. (2023a); Liu et al. (2023c); Zhang et al. (2023); Lin et al. (2023); Laurençon et al. (2024) have showcased powerful comprehension abilities in understanding multimodal data, which makes them ideally suited for interleaved image-text content in multimodal stories. Consequently, we introduce SEED-Story, as shown in Figure 1, a novel approach that builds upon the MLLM to harness its comprehension strength, while further equipping it with the capability to generate coherent images align with the narrative texts.

Specifically, following previous work Sun et al. (2023a); Ge et al. (2024), we utilize a pre-trained image tokenizer and de-tokenizer, which can decode realistic images with SD-XL Podell et al. (2023) by taking the features of a pre-trained ViT as input. During training, given the interleaved visual and textual data, we adopt the next-word prediction and image feature regression training objectives to regularize multimodal generation. A fixed number of learnable queries are fed into the MLLM, where the output hidden states are trained to reconstruct the ViT features of the target images. To further enhance the consistency of characters and styles in generated images, we propose de-tokenizer adaptation, where the regressed image features from the MLLM are fed into the de-tokenizer for tuning SD-XL. This adaptation allows for better maintenance of coherence in low-level image details from the de-tokenizer, ensuring a more visually consistent storytelling output.

Furthermore, to enable the efficient generation of coherent long stories, we propose a multimodal attention sink mechanism based on window attention Beltagy et al. (2020), which maintains a fixed-size sliding window on the Key-Value (KV) states of the most recent tokens, as well as the beginning of text tokens, images tokens, and the end of image tokens. We empirically find that retaining these

tokens will largely mitigate the model's failure with window attention when the token length surpasses the cache size, allowing our model to generalize to longer sequences than the training sequence length in an efficient manner. Our model with the proposed multimodal attention sink mechanism can generate long stories with up to 30 multimodal sequences, featuring rich text plots and diverse visual scenarios.

Additionally, we introduce a dataset named StoryStream for training and evaluating multimodal story generation. We design an automatic pipeline that leverages MLLMs to obtain a large-scale and high-resolution dataset featuring a sequence of narrative-rich texts and intriguing images, derived from animated videos. StoryStream is four times larger in terms of data volume compared to the existing largest story dataset Liu et al. (2023a), and it boasts higher image resolution, longer sequence lengths, and more detailed story narratives. We further meticulously design evaluation metrics to assess multimodal story generation, taking into account image style consistency, story engagement, and image-text coherence. The evaluation results demonstrate that our model, SEED-Story, achieves superior performance in these aspects.

In summary, Our contributions are three-fold. (1) We propose SEED-Story, a novel method that leverages an MLLM to generate multimodal stories with rich narrative text and contextually relevant images. (2) We propose multimodal attention sink to enable the efficient generation of long stories with sequence lengths larger than those used during training. (3) We introduce StoryStream, a large-scale dataset specifically designed for training and benchmarking multimodal story generation. All models, training and inference codes are released in `https://anonymous.4open.science/r/SEED-Story/`.

## 2 RELATED WORK

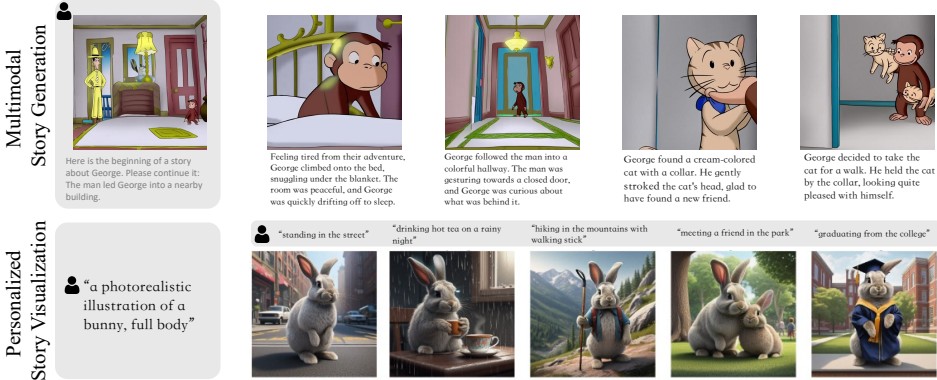

Figure 2: Comparison between personalized story visualization and multimodal story generation. Contents in grey boxes refer to user's input. In the former, a sequence of captions are given (referred to as a "story") for consistent text-to-image generation. By contrast, multimodal story generation involves creating a sequence of consecutive storylines along with their corresponding images.

**Personalized Story Visualization v.s. Multimodal Story Generation**  Personalized Story Visualization, such as Li et al. (2019); Maharana et al. (2021); Maharana & Bansal (2021); Maharana et al. (2022); Pan et al. (2024); Rahman et al. (2023); Gong et al. (2023); Liu et al. (2023a); Ruiz et al. (2023); Avrahami et al. (2024); Tewel et al. (2024), aims to generate images depicting specified characters engaged in various actions or within different scenes, based on **the provided captions** (so-called "story") as shown in Fig. 2, which follows the pattern of **text-to-image generation**. For example, StoryDALL-E Maharana et al. (2022) utilizes pre-trained models augmented with cross-attention layers to support story progression from an initial image. Innovations like AR-LDM Pan et al. (2024) and Story-LDM Rahman et al. (2023) have introduced auto-regressive diffusion models to create coherent sequences of images, while TaleCrafter Gong et al. (2023) has employed LoRA and optimization techniques to ensure consistent characters throughout complex visual narratives.

Table 1: Comparison of multimodal story generation datasets. The table provides details on the number of images, their resolution, the total length of visual stories, and the average text length per sentence, which indicates the narrative detail of the text. Note that StorySalon has various size of images and we choose one of the typical sizes presented here.

| Datasets | Number of Images | Resolution | Story Length | Avg Text Length |
|---|---|---|---|---|
| Flintstones Gupta et al. (2018) | 122,560 | $128 \times 128$ | 5 | 86 |
| Pororo Li et al. (2019) | 73,665 | $128 \times 128$ | 5 | 74 |
| StorySalon Liu et al. (2023a) | 159,778 | $432 \times 803$ | 14 | 106 |
| **StoryStream** | 257,850 | $480 \times 854$ | 30 | 146 |

Multimodal Story Generation (Shen & Elhoseiny (2023)) aims to generate **a sequence of consecutive storylines along with their corresponding images**, similar to that of a serialized comic. To achieve this, a model must be capable of predicting reasonable story developments and generating corresponding illustrations, by incorporating the previous results as context in an auto-regressive manner. As shown in Figure 2, the generated story images exhibit rapidly changing backgrounds and characters, different from personalized story visualization where the main character consistently appears in the images.

The task of multimodal story generation presents a more substantial challenge, and we follow previous research Shen & Elhoseiny (2023) to adopt a closed-set setting. We believe the ultimate goal of multimodal story generation should be to generate highly diverse scenarios while also generalizing to unseen characters, which will be explored in our future work.

**MLLM for Multimodal Story Generation**   In the rapidly evolving domain of large language models (LLMs) Touvron et al. (2023); Brown et al. (2020); Chowdhery et al. (2022) and multimodal large language models (MLLMs) Li et al. (2023); Zhu et al. (2023a); Liu et al. (2023d); Peng et al. (2023b); Bai et al. (2023b); Liu et al. (2023b); Zhang et al. (2023); Lin et al. (2023); Sun et al. (2023c); Yu et al. (2023); Ge et al. (2023c;b); Wu et al. (2023); Dong et al. (2023); Zhu et al. (2023b); Sun et al. (2023b); Li et al. (2024), recent work StoryGPTV Shen & Elhoseiny (2023) explores using MLLMs for story generation by converting visual features into token embeddings for image generation, but requires aditional character and object masks for training. MM-interleaved Tian et al. (2024) designs a multi-scale and multi-image feature synchronizer module (MMFS) to process interleaved text-image data and generates next image conditioned on the previous context features from LLM, which makes it difficult to generate long multimodal stories due to the complex multi-scale attention mechanism.

**Visual Story Dataset**   In the landscape of datasets for visual storytelling, various collections have been developed as shown in Table 1. The VIST Huang et al. (2016) dataset is noteworthy for its use of realistic images, though it struggles with maintaining character consistency across stories. Pororo Li et al. (2019) and Flintstones Gupta et al. (2018) datasets, while popular for animation-based story datasets, are hindered by their low resolution and the simplicity of their accompanying texts. Another significant dataset is StorySalon Liu et al. (2023a), which offers high-resolution images and is large in scale, but it lacks global consistency across images. To address these gaps, we introduce StoryStream, a globally consistent, large-scale, high-resolution animated style dataset with engaging, narrative-rich text for complex storytelling, overcoming the limitations of existing datasets.

## 3 METHOD

### 3.1 STORY GENERATION WITH MULTIMODAL LARGE LANGUAGE MODEL

**Visual Tokenization and De-tokenization**   The overview of our method is presented in Figure 3. To effectively extend visual stories, our model must comprehend and generate both images and text. Drawing inspiration from recent advancements in generative MLLMs that unify image comprehension and generation Podell et al. (2023), we develop a multimodal story generation model. Our model employs a pre-trained Vision Transformer Dosovitskiy et al. (2020) (ViT) as the visual tokenizer and a pre-trained diffusion model as the visual de-tokenizer to decode images by using ViT's features as inputs. Specifically, visual embeddings from the ViT tokenizer are fed into a learnable module, which

Figure 3: Overview of the SEED-Story Training Pipeline: In **Stage 1**, we pre-trains an SD-XL-based de-tokenizer to reconstruct images by taking the features of a pre-trained ViT as inputs. In **Stage 2**, we sample an interleaved image-text sequence of a random length and train the MLLM by performing next-word prediction and image feature regression between the output hidden states of the learnable queries and ViT features of the target image. In **Stage 3**, the regressed image features from the MLLM are fed into the de-tokenizer for tuning SD-XL, enhancing the consistency of the characters and styles in the generated images.

then serves as inputs for the U-Net of the pre-trained SD-XL Podell et al. (2023). This process replaces the original text features with visual embeddings. During this stage, the parameters are optimized using open-world text-image pair data as well as story data to enhance the model's encoding-decoding capability. After this training phase, we expect the visual tokenizer and de-tokenizer modules to preserve as much image information as possible in the feature space.

**Story Instruction Tuning** In our instruction tuning process for story generation, we sample a random-length subset of a story data point for each iteration. The model is tasked with predicting the next image and the next sentence of the story text. Within MLLM, all images are converted into image features using a pre-trained ViT tokenizer. For the target text tokens, we perform next-token prediction and use Cross Entropy loss to train for this discrete target. For target image features, the model uses a series of learnable queries as inputs and continuously outputs a series of latents. We then compute the cosine similarity loss between the MLLM's output and the target image features. During this stage, we fine-tune the SEED-Story model using a LoRA Hu et al. (2021) module.

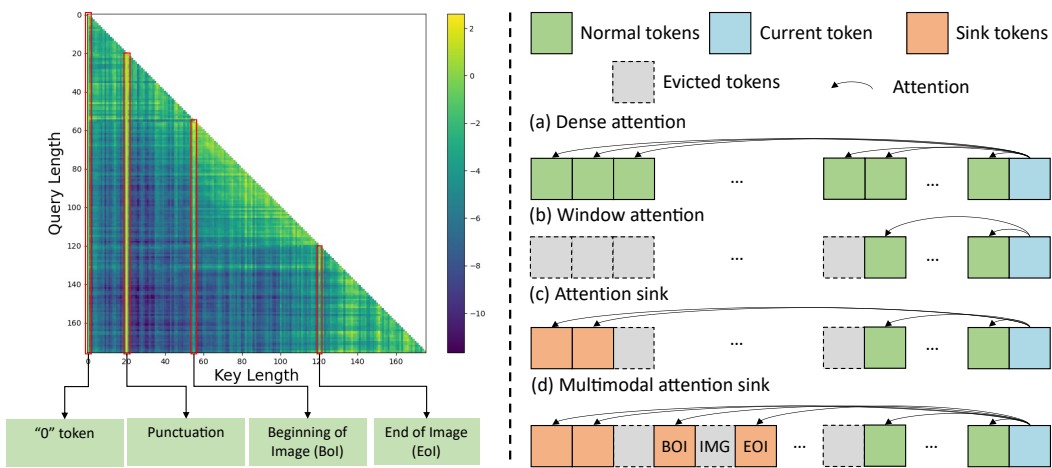

Figure 4: **Left:** Visualization of the attention map when predicting the next token for multimodal story generation. We observe that important attentions are aggregated into the first token of the whole sequence ("0" token), punctuation tokens, tokens adjacent to BoI, and tokens adjacent to EoI. **Right:** The diagram of (a) Dense attention, which preserves all tokens in KV cache. (b) Window attention, which evicts preceding tokens by a sliding window. (c) Attention sink, which preserves the beginning tokens based on window attention. (d) Multimodal attention sink, which preserves the beginning of text tokens, images tokens, and the end of image token based on window attention. It can efficiently enable our model to generalize to generating longer sequences than the training sequence length.

**De-tokenizer Adaptation** After instruction tuning, the SEED-Story MLLM effectively produces story images with correct semantics but lacks style consistency and details. We attribute this issue to the misalignment between the latent space of the MLLM output and the image features. To address this, we perform de-tokenizer adaptation for style and texture alignment. In this stage, only the SD-XL image de-tokenizer is trained. Conditioned on the MLLM output embeddings, SD-XL is expected to generate images that are pixel-level aligned with the ground truth. The separate training of the de-tokenizer offers two key advantages. First, it avoids optimization conflicts between the LLM and the de-tokenizer. Second, it conserves memory, making the process executable on GPUs with limited memory. Please see more analysis in Section B of our appendix.

## 3.2 LONG STORY GENERATION WITH MULTIMODAL ATTENTION SINK

Generating long visual stories has substantial potential in various applications, including education and entertainment. However, creating these stories with MLLMs presents significant challenges. Datasets for extended, interleaved stories are not only rare but also impede the training process due to their complexity. To address this, we have to employ a train-short-test-long approach, training models on shorter narratives and extending to longer generations during inference.

Moreover, during inference, generating significantly longer stories than the training data often leads to model degradation, producing lower-quality images, as illustrated in the first row of Figure 8. This process also requires extensive token usage to ensure continuity and coherence, which in turn increases memory and computational demands.

A simplistic solution for this is to use a sliding window technique, depicted in Figure 4 right (b). However, this method disrupts the token relationships in the Key-Value (KV) cache, resulting in subpar generative outcomes, as demonstrated by StreamingLLM Xiao et al. (2023). To overcome this, StreamingLLM introduces an attention sink mechanism that preserves the initial tokens, thus allowing for efficient processing of lengthy generations without quality compromise. While effective in language models, its efficacy diminishes in multimodal contexts, as shown in Figure 4 right (c).

To enhance long multimodal generation, we revisit the attention maps of MLLMs. After conducting numerous experiments (see Section D of the appendix for more details) across various models and cases, we analyze the attention maps across different layers and heads. Our analysis reveals that most

queries predominantly focus on four types of tokens: (1) starting tokens, (2) punctuation tokens, (3) beginning-of-image (BoI) tokens, and (4) end-of-image (EoI) tokens. Unlike language-only models, MLLMs place considerable attention on specific image tokens, particularly those near the BoI and EoI, as illustrated in Figure 4 left.

Building on these insights, we propose a new mechanism for extended generation in MLLMs, termed the multimodal attention sink. During generation, we consistently retain the starting tokens and the image tokens adjacent to the BoI and EoI. Although punctuation tokens receive high attention values, their latent value norms are minimal, contributing insignificantly to the final output, so we do not keep them, as noted by Ge et al. (2023a). Our proposed mechanism enables our model to generate high-quality images while maintaining a low computational footprint.

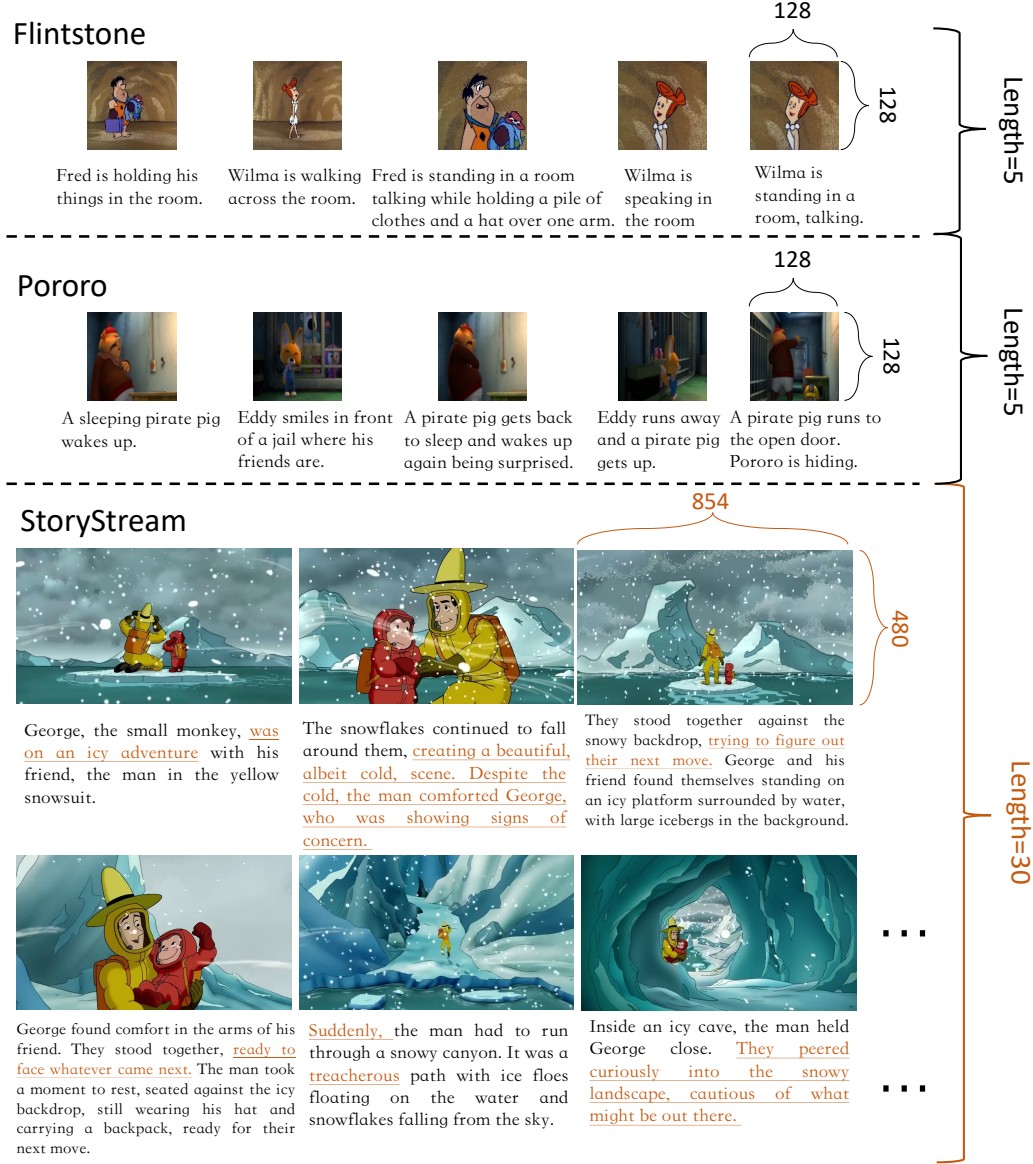

Figure 5: Data sample of our StoryStream dataset and existing multimodal story generation datasets. Our multimodal story sequences consist of high-resolution images that are visually engaging, and detailed narrative texts as underlined, closely resembling the real-world storybooks. Additionally, our stories are more extended in length.

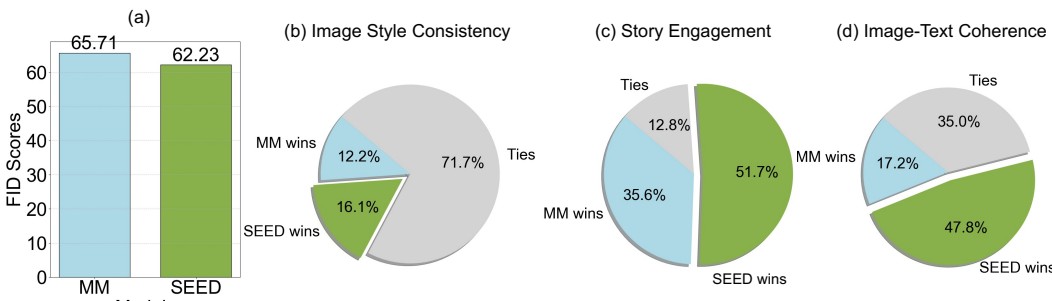

Figure 6: Quantitative evaluation of multimodal story generation between MM-interleaved versus SEED-Story. **(a):** Histograms displaying FID scores. **(b, c, d):** GPT-4V is employed to choose a preferred result generated by MM-interleaved and SEED-Story respectively. Pie charts show the win rate, where "Ties" indicates GPT-4V assesses their outcomes with equal scores.

# 4 STORYSTREAM DATASET

## 4.1 DATASET CONSTRUCTION

An ideal source for creating a multimodal story generation dataset is cartoon series, which inherently contain rich plots and consistent character portrayals. We selected three cartoon series to construct our dataset and we present the Curious George in the main body of our paper. The process begins with collecting various series, from which we extract keyframes and their associated subtitles Kilian et al. (2023). Each keyframe is then processed by GPT-4V OpenAI (2023) or Qwen-VL Bai et al. (2023b) to generate a detailed image description. These elements—keyframe, subtitle, and description—are compiled into a single group. We aggregate 30 such groups and input them into GPT-4, supplemented with background information about the cartoon series. Following our instructions, GPT-4 generates high-quality narrative texts suitable for training story generation models.

During dataset construction, we discovered that employing the above chain of thought approach not only produces more accurate narrative text but also speeds up the construction process. Unlike directly feeding all images directly to GPT-4, which is limited to 10 images due to API constrains, our approach produces longer stories. We also significantly improve the model's understanding of each image by incorporating detailed descriptions. This enhancement in image comprehension enriches the narrative details, providing a richer story generation reference.

## 4.2 KEY FEATURES

**Large-scale.** Our StoryStream dataset comprises three subsets totaling 257,850 released images. This represents a significant improvement over existing datasets in terms of scale, specific numbers are presented in column 2 of Table 1. To the best of my knowledge, it is the largest visual story generation dataset featuring consistent main characters.

**High Resolution.** Unlike existing story generation datasets which offer images at a resolution of 128x128, our story stream dataset provides high-resolution images of 480x768.

**Narrative Text.** Our dataset diverges from existing ones that utilize simple and descriptive language. We offer abstract, narrative, detailed, and story-toned texts that are more akin to real-world applications, such as visualizing narratives from a storybook, examples are shown in Figure 5. Story text of existing datasets obey the form of "name" + "action", like "Poby is playing the violin.". Contrarily, our story text involves more intrinsic elements. This effectively enhances the engagement of audiences. An analysis of the average text length per sentence is shown in column 5 of Table 1.

**Long Sequence.** Moreover, our dataset enhances long story comprehension by offering up to 30 images per story point. Within these 30 images, our corresponding texts present a cohesive narrative, effectively conveying the progression and intricacies of extended stories.

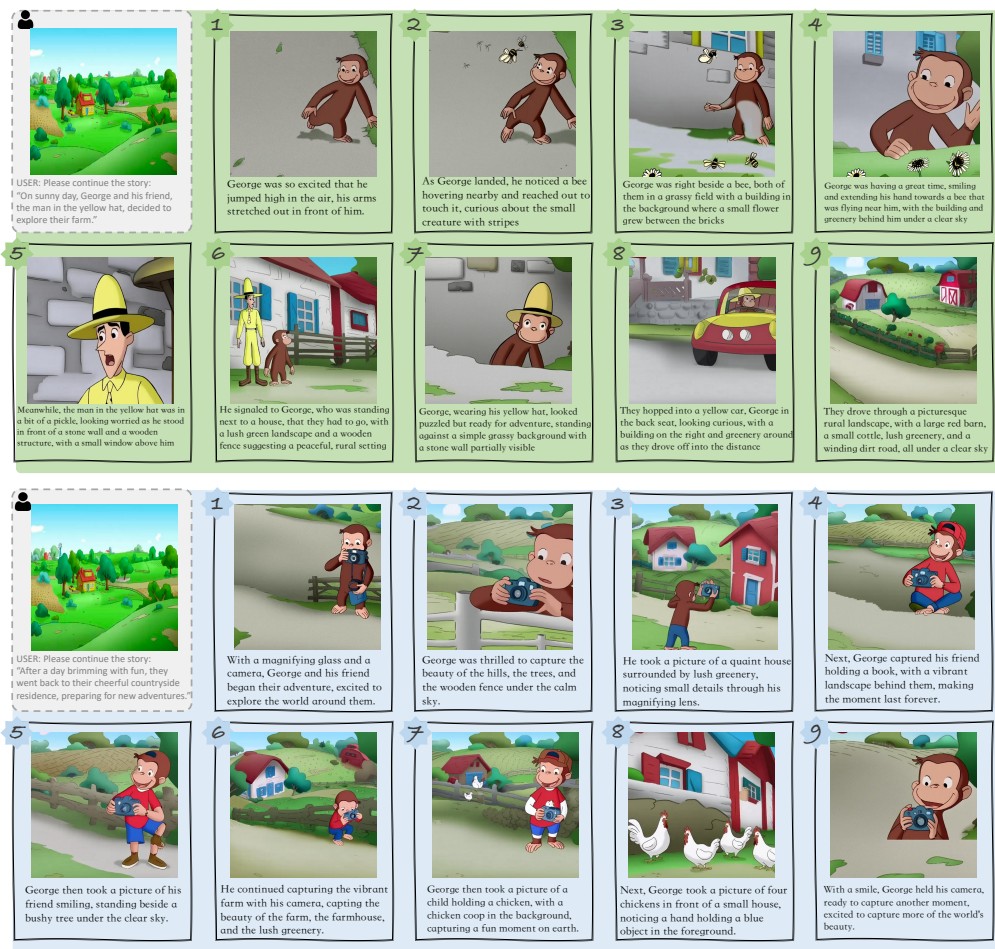

Figure 7: Examples of multimodal story generation from SEED-Story with the same initial image but different texts as the beginning. The top branch starts with the text referencing "the man in the yellow hat", which leads to the generation of images that include the character. The bottom branch starts without mentioning the man, resulting in a different storyline.

## 5 EXPERIMENT

### 5.1 QUANTITATIVE RESULTS

For quantitative evaluation of multimodal story generation, since there are relatively few established methods for generating multimodal stories, we first fine-tune the recently developed MM-interleaved model on our training dataset for a fair comparison as the baseline model. The quantitative results are listed in Figure 6. The FID is employed to assess the visual quality of the generated images. Additionally, we ask GPT-4V ("gpt-4-turbo-2024-04-09") to compare and choose a preferred option between each of the generation results of MM-interleaved and SEED-Story across several dimensions: (a) Style Consistency, which evaluates the stylistic uniformity across different images; (b) Story Engagement, which measures the ability of narratives to captivate and maintain audience interest; (c) Image-Text Coherence, which assesses the alignment and relevance between images and their accompanying texts. The details of evaluation are introduced in Section G of Appendix. We also conduct a user study, which compares SEED-Story and MM-Interleaved in Section A of Appendix. The quantitative evaluations demonstrate that SEED-Story outperforms the baseline model in terms of story engagement and image-text coherence, and achieves slightly higher preference in image style consistency. We also provide quantitative comparisons of story visualization in Section E of Appendix.

Figure 8: The visualization of generating long stories with different attention mechanisms. Without multimodal attention sink, MLLM cannot generate meaningful long image sequences. As highlighted in the red boxes, other methods produce meaningless images in the later frames.

Table 2: Quantitative evaluation of long story generation with various attention mechanisms. FID and CLIP scores are calculated by comparing ground truth images with generated images. Inference time and memory usage are calculated by generating 50 sequences multiple times for average.

| Metrics | FID ↓ | CLIP Score ↑ | Inference Time (s) ↓ | Memory (GB) ↓ |
|---|---|---|---|---|
| **Dense Attn** | 119.72 | 0.705 | 569.67 | 37.99 |
| **Window Attn** | 334.90 | 0.598 | **450.64** | **30.81** |
| **Attn Sink** | 221.53 | 0.676 | 451.94 | **30.81** |
| **Multimodal Attn Sink** | **79.67** | **0.728** | 473.98 | 31.82 |

## 5.2 QUALITATIVE RESULTS

For qualitative evaluations, we first demonstrate that our SEED-story can effectively generate stories with different plots and corresponding illustrations based on the different beginnings provided by the user, as shown in Fig. 7. We provide more qualitative results of multimodal story generation in Section F of Appendix. As shown in Figure F, Figure G and Figure H, SEED-story can generate long sequences with engaging plots and vivid images. We further provide qualitative comparisons of story visualization in Section E of Appendix.

## 5.3 MULTIMODAL ATTENTION SINK

To verify the effectiveness of multimodal attention sink in long story generation, we conduct an experiment visualizing a long story using the SEED-Story model, but with varying attention mechanisms. We chunk our data into stories of length of 10 considering the training efficiency. We set the window size as the same as the training length. Qualitative results presented in Figure 8 demonstrate that window attention quickly collapses when the inference length exceeds the training length. Both dense attention and attention sink approaches fare better, yet still fail to produce meaningful images as the inference sequence lengthens. In contrast, the multimodal attention sink consistently produces high-quality images. In terms of efficiency, the multimodal attention sink exhibits significant improvement over dense attention, with only a modest increase in time and memory costs compared to window attention and vanilla attention sink. These additional costs stem from retaining extra image tokens in the KV cache. Quantitative results presented in Table 2 substantiate the above conclusion.

## 6 CONCLUSION

This work introduces SEED-Story, a pioneering approach that leverages a Multimodal Large Language Model to generate multimodal long stories with rich narrative text and contextually relevant images. We propose a multimodal attention sink mechanism to enable our model to generalize to generating long sequences in an efficient manner. We further present a high-quality dataset named StoryStream for training and benchmarking the task of multimodal story generation effectively.

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

APPENDIX

## A  USER STUDY

| Criteria | MM-Interleaved | SEED-Story | Ties |
|---|---|---|---|
| Image Quality | 21.3% | 66.2% | 12.5% |
| Image Style Consistency | 7.5% | 78.8% | 13.7% |
| Image Diversity | 21.3% | 75.0% | 3.7% |
| Story Engagement | 37.5% | 55.0% | 7.5% |
| Image-Story Coherence | 5.0% | 86.3% | 8.7% |

Table 3: The results of user study between MM-Interleaved, SEED-Story, and Ties

Participants were asked to choose their preference based on image quality, image style consistency, image diversity, story engagement, and image-story coherence. The results were obtained from 125 samples, as shown in Table 3.

The results indicate that SEED-Story clearly outperforms the baseline in image generation ability and text-to-image coherence. Additionally, SEED-Story shows a slightly higher preference in text quality for story generation.

## B  ABLATION STUDY ON DE-TOKENIZER ADAPTATION

We find that the generated images before the de-tokenizer adaptation stage exhibit semantic relevance with consistent backgrounds and characters, thanks to MLLM's context preservation, as shown in Figure A. However, they suffer from texture distortion and inconsistency in style. After de-tokenizer adaptation, the images show improved consistency in style and character appearance. The calculated FID scores in table 4 confirm that de-tokenizer adaptation enhances image quality.

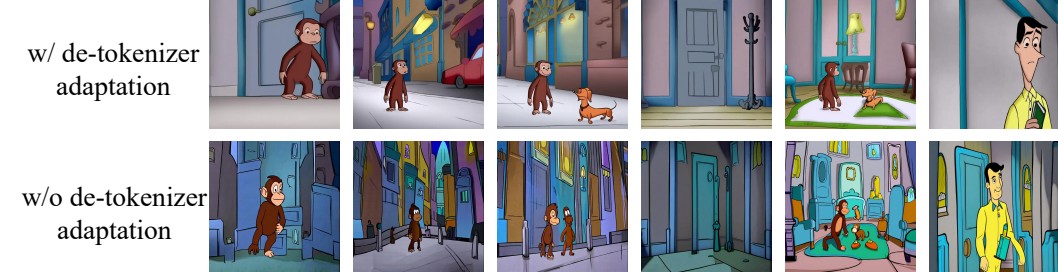

Figure A: Generated story images with and without the 3rd stage: **de-tokenizer adaptation**. The images generated before the third stage preserve semantic information, with mostly correct backgrounds and characters. However, they display low-quality textures and inconsistency in cartoon style. Our third stage effectively enhances these aspects.

| Model | FID |
|---|---|
| Before 3rd stage | 153.93 |
| After 3rd stage | 99.79 |

Table 4: FID scores before and after the 3rd stage.

## C  IMPLEMENTATION DETAILS

### C.1  VISUAL TOKENIZATION AND DE-TOKENIZATION

For visual tokenization, we use Qwen-VL pre-trained ViT-G. We first resize the image to 448x448 images and then use ViT to produce its feature of length256 with 4096 dimension. (shape: [256,

4096]). Inside the MLLM, we use a Q-Former architecture to process the image embedding. It takse the ViT image feature as key and Value, and conduct attention with its learnable queries. The length of learnable queries are 64. For de-tokenization, we also use a Q-Former architecture to transform the MLLM output to the shape of SD-XL condition embedding.

## C.2 INSTRUCTION TUNING

Instruction tuning data is formatted as follows: for each story, we sample a random length and compute losses on the last sequence (highlighted in red text). The sequence format is structured as:

```
<bos>[start of the story.][User prompt:  ][following
sequence 1][following sequence 2][following sequence
3][following sequence 4] ... [target sequence]<eos>
```

For our language model (LLM), we utilize the LLAMA2-7B pre-trained model and finetune it using LoRA, supported by the *peft* library. The hyperparameter $r$ is set to 6, and lora_alpha is set to 32. The modules optimized include the $q\_projection\_layer$, $v\_projection\_layer$, $k\_projection\_layer$, $o\_projection\_layer$, $gate\_projection\_layer$, $down\_projection\_layer$, and $up\_projection\_layer$. We employ a learning rate of $1 \times 10^{-4}$ to finetune this model on our dataset across approximately 6 epochs, utilizing 8 NVIDIA-A800 GPUs.

## C.3 DE-TOKENIZER ADAPTATION

In this stage we fully finetune the SD-XL model. The data format is as the same as instruction tuning, but we fix all MLLM params and optimize only the SD-XL. It takes the MLLM output and is asked to produce image correspond to the ground truth. The SD-XL model was trained using 4 NVIDIA-A800 GPUs. A learning rate of $1 \times 10^{-4}$ was chosen to facilitate gradual weight updates, ensuring stable convergence, while a weight decay of 0.03 was applied for regularization to prevent overfitting. Training was performed using mixed precision (`bf16`), which significantly reduced memory usage and accelerated the training process without compromising the model's accuracy. The model underwent three training epochs, balancing the learning of complex patterns against computational resource use, optimized for large-scale datasets and sophisticated model architectures.

# D ANALYSIS OF MULTIMODAL ATTENTION SINK

## D.1 ATTENTION MAP VISUALIZATION

In this section, we present additional visualizations of attention maps. These maps are derived from various model runs, including varying data lengths, attention heads, and layers. The visualizations consistently reveal a pattern of attention focused on "0" tokens, punctuation, tokens adjacent to Begin-of-Image (BoI), and tokens adjacent to End-of-Image (EoI).

## D.2 STATISTICS ON TOKEN'S ATTENTION

To demonstrate that more attention is paid to image tokens adjacent to **<BOI>** and **<EOI>**, we analyzed over 5600 attention maps from various models, layers, and input sequences, to identify "key tokens" with the highest mean attention values. For each attention map, we computed the mean attention value across the query dimension for every key and selected the top 10 keys. We then aggregated these results to determine how often each token appeared as a key token. Table 5 shows the tokens with the highest occurrences, supporting our multimodal attention sink mechanism, with most queries focusing on four key token categories.

- Starting tokens (BOS)
- Punctuation ("," "." and ";" ...)
- Image tokens near BOI (BOI, IMG04)
- Image tokens near EOI (from IMG57 to IMG63, added EOI)

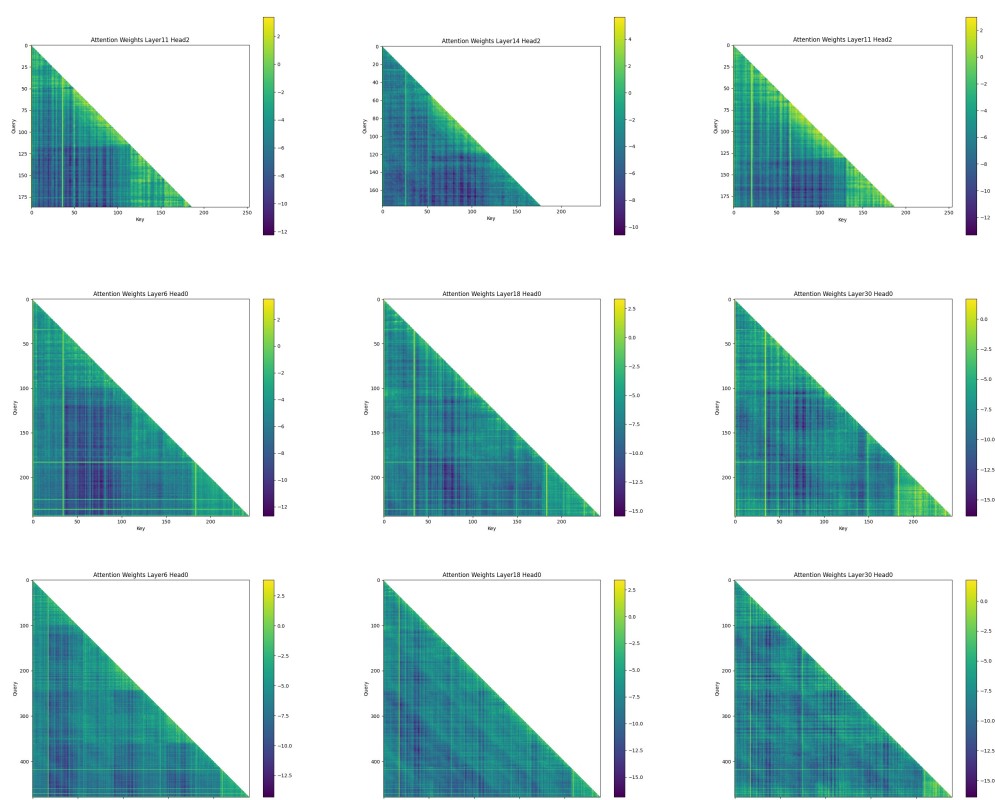

Figure B: Visualization of attention maps from various model runs, showcasing attention patterns across different data lengths, attention heads, and model layers. Notably, the maps highlight consistent focus on '0' tokens, punctuation, tokens adjacent to Begin-of-Image (BoI), and tokens adjacent to End-of-Image (EoI).

|  | BOS | IMG57 | EOI | IMG04 | , | IMG60 |
|---|---|---|---|---|---|---|
| **Key Token Occurrence** | 4320 | 4320 | 4320 | 4140 | 4140 | 4120 |
|  | **IMG61** | **IMG59** | **IMG62** | **IMG63** | **BOI** | **IMG56** |
| **Key Token Occurrence** | 3730 | 3132 | 1651 | 607 | 603 | 361 |

Table 5: Key Token Occurrence for Various Tokens

# E   STORY VISUALIZATION COMPARISON

Previous story generation approaches primarily utilize diffusion models, focusing on visualizing story images. These models take the previous image and text as input, and then generate only the next image based on the current text prompt. For a fair comparison, we adapt our model to a visualization-only format. For StoryGen Liu et al. (2023a), we also train it to produce images with previous images and texts. For LDM Rombach et al. (2022), we

Table 6: Quantitative evaluation for story visualization.

| Model | FID ↓ | CLIP Score ↑ |
|---|---|---|
| LDM | 67.29 | 0.7585 |
| StoryGen | 73.74 | 0.7573 |
| **SEED-Story** | **67.01** | **0.7793** |

only give it text-image pairs. The visual results are shown in Figure C. SEED-Story model shows better style and character consistency and higher quality compared to baselines. We also showcase the visualization result of our model on Rabbids Invasion and The Land Before Time. Please see Figure D and E. We further conduct a quantitative evaluation in Table 6 to demonstrate our effectiveness.

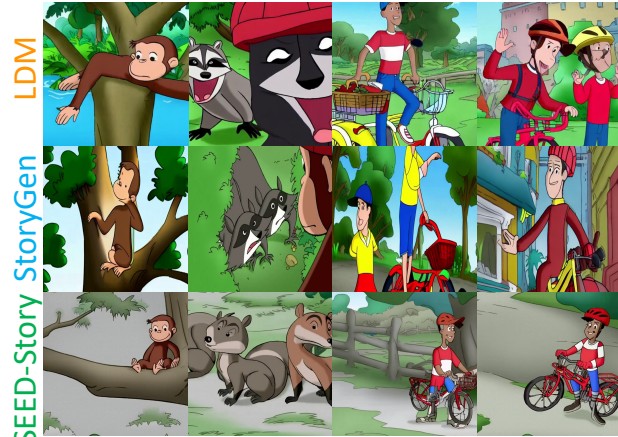

a) And indeed, it had! A small, brown monkey named George was sitting on a nearby tree branch, hugging his legs with a curious expression.

b) To his surprise, two raccoons, with mouths open and startled expressions, were looking at George. He was only partially visible, with his red hat peeking out from the bush.

c) Suddenly, a character wearing a red and white shirt, blue pants, and a yellow cap appeared. He stood a red bike with a basket, in the lush green park with trees and a wooden fence.

d) The character stood next to his red and yellow bicycle in the green park. He raised a hand to wave at George.

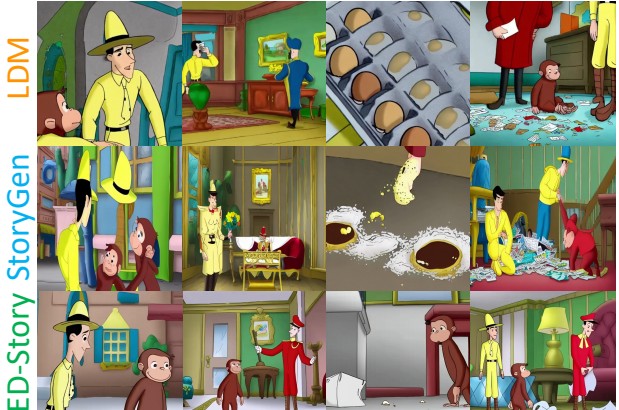

a) Suddenly, the man in the yellow hat stopped to look at something off-screen. George, the curious monkey, also turned to look in the same direction.

b) The bellhop continued his conversation on the phone while another figure held wooden planks. The room was filled with a classic armchair, console table with a flower vase, and a framed painting.

c) Suddenly, there was a spill! An eggs had fallen, and cracked eggs, yolk, and eggshells were scattered.

d) George sat on the floor with the man in the yellow suit and the bellhop in the red coat, amidst scattered pieces of paper and a broken object.

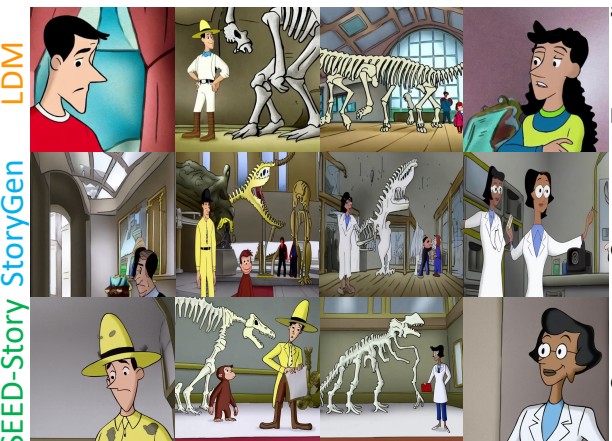

a) The man looked to the side with a surprised expression. The background appeared to be the interior of a rounded structure with a window.

b) A small monkey looked up at a person wearing an outfit. In the background, another person stood next to a large dinosaur skeleton inside what appeared to be a museum.

c) A dinosaur skeleton stood inside a museum, with a woman in a coat gesturing towards it. In the background, a small figure waved enthusiastically from behind a glass window.

d) An woman with short hair and a lab coat stood with hands on hips, smirking. The background showed a wall with two light switches.

Figure C: Story visualization comparison of SEED-Story and other story visualization methods.

## F  MULTIMODAL STORY GENERATION RESULTS

In this section, we present more multimodal story generation results of our SEED-Story. It keeps produce story image and text with high quality. Figure F, Figure G, and Figure H prove our multimodal

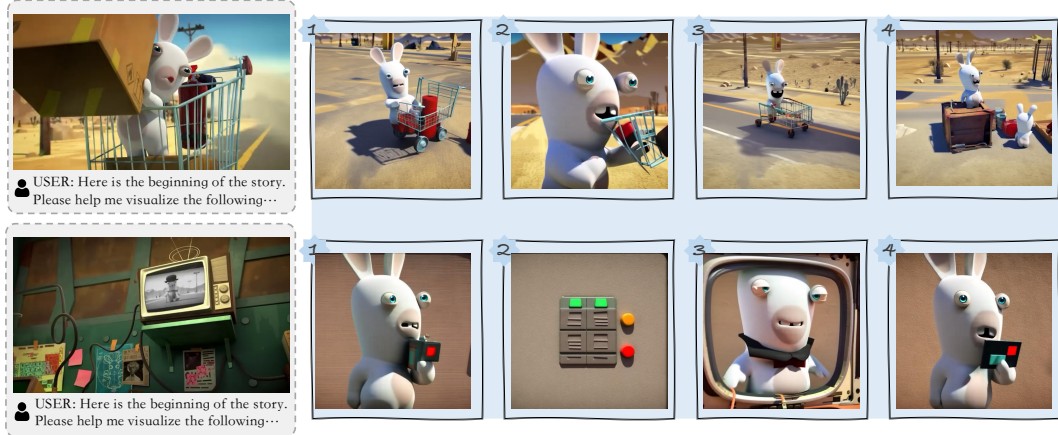

Figure D: Story visualization result on Rabbids Invasion data.

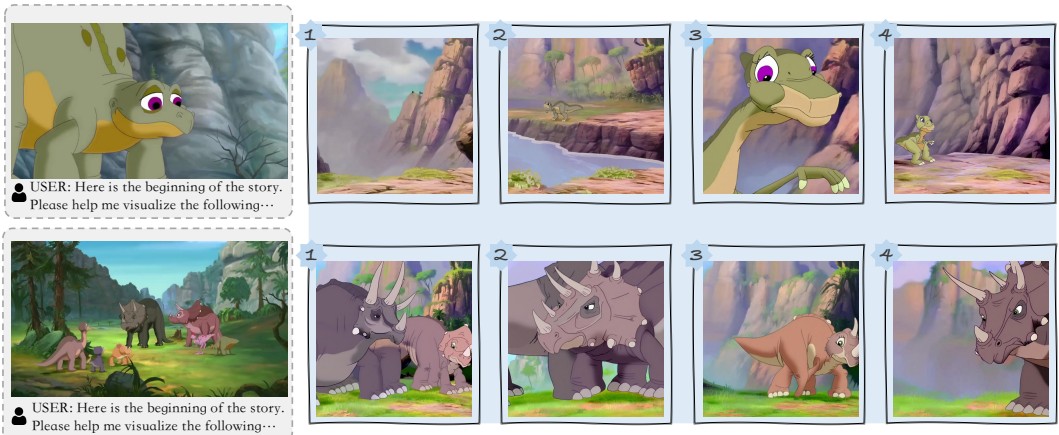

Figure E: Story visualization result on The Land Before Time data.

long story generation capabilities. SEED-story can generate long sequences with engaging plots and vivid images.

# G DETAILS ABOUT GPT-4V EVALUATION

## G.1 COMPARATIVE EVALUATION

To evaluate the effectiveness of MM-interleaved and SEED-Story in multimodal story generation, we initiate an experiment where each model produces a story of five segments, based on a common starting image and text. The segment limit is set to five to accommodate the constraints of GPT-4V, which can handle a maximum of ten images per input session. In total, we generate 180 stories for assessment. For evaluation, we employ GPT-4 or GPT-4V to determine which model produces the better story in each case, based on the framework established in L-Eval An et al. (2023). We calculate the win rate for each model to determine its performance relative to its counterpart. The prompt we used is shown below.

> "Please act as an impartial judge and evaluate the quality of the generation story contents provided by two AI assistants. Your job is to evaluate which assistant's generation is better. Your evaluation should consider **{the style consistency of the**

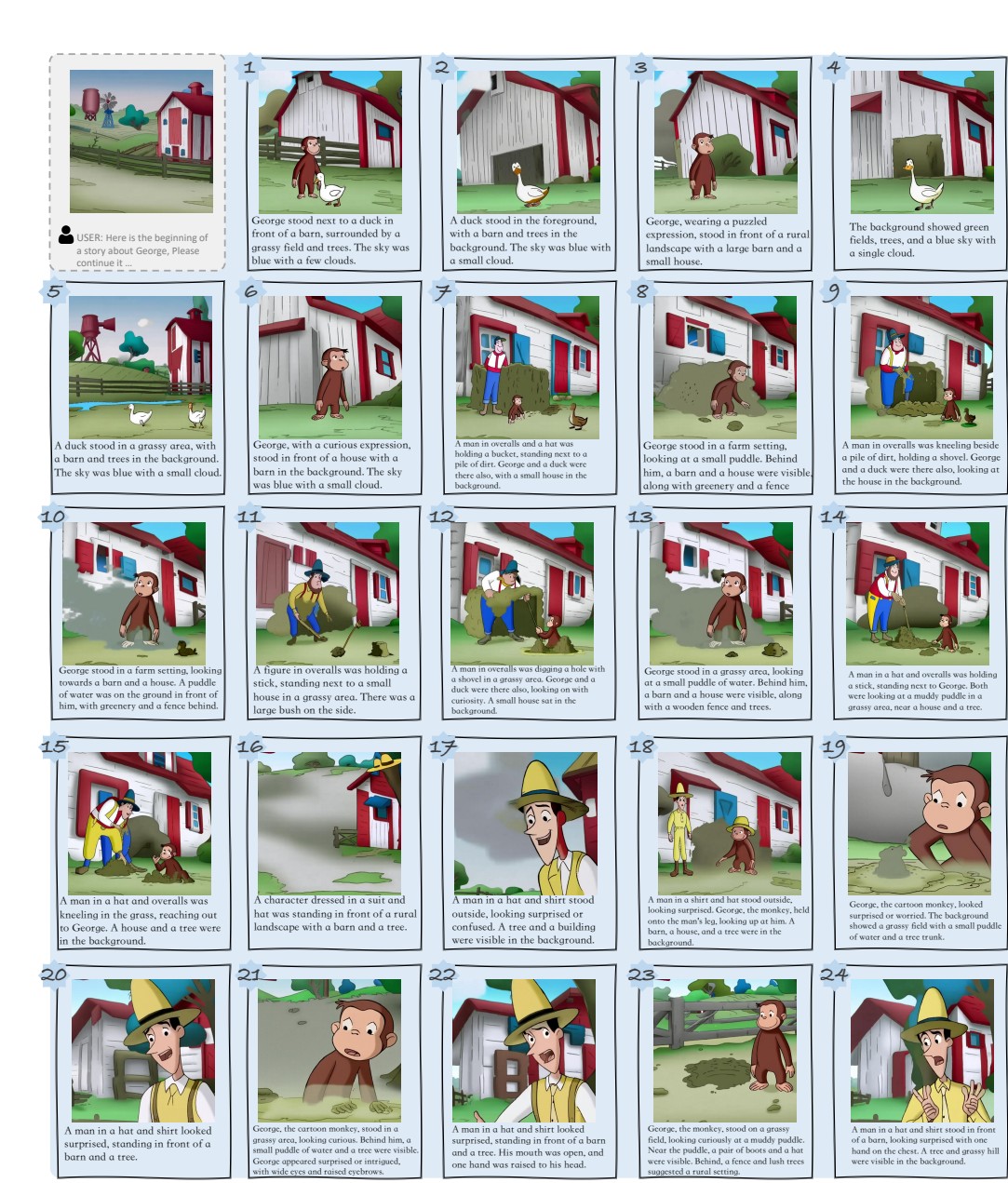

Figure F: Multimodal long story generation results of SEED-Story.

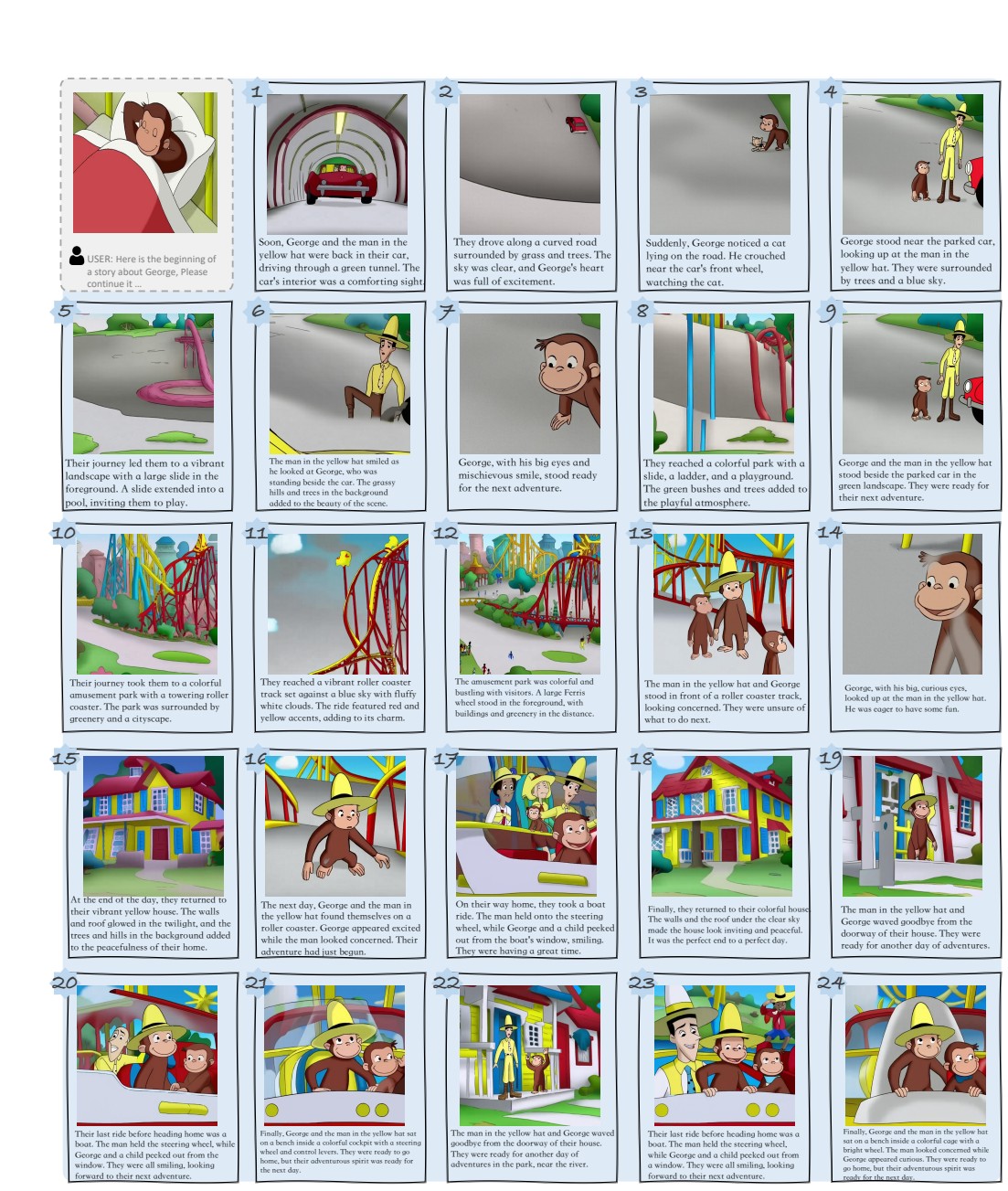

Figure G: Multimodal long story generation results of SEED-Story.

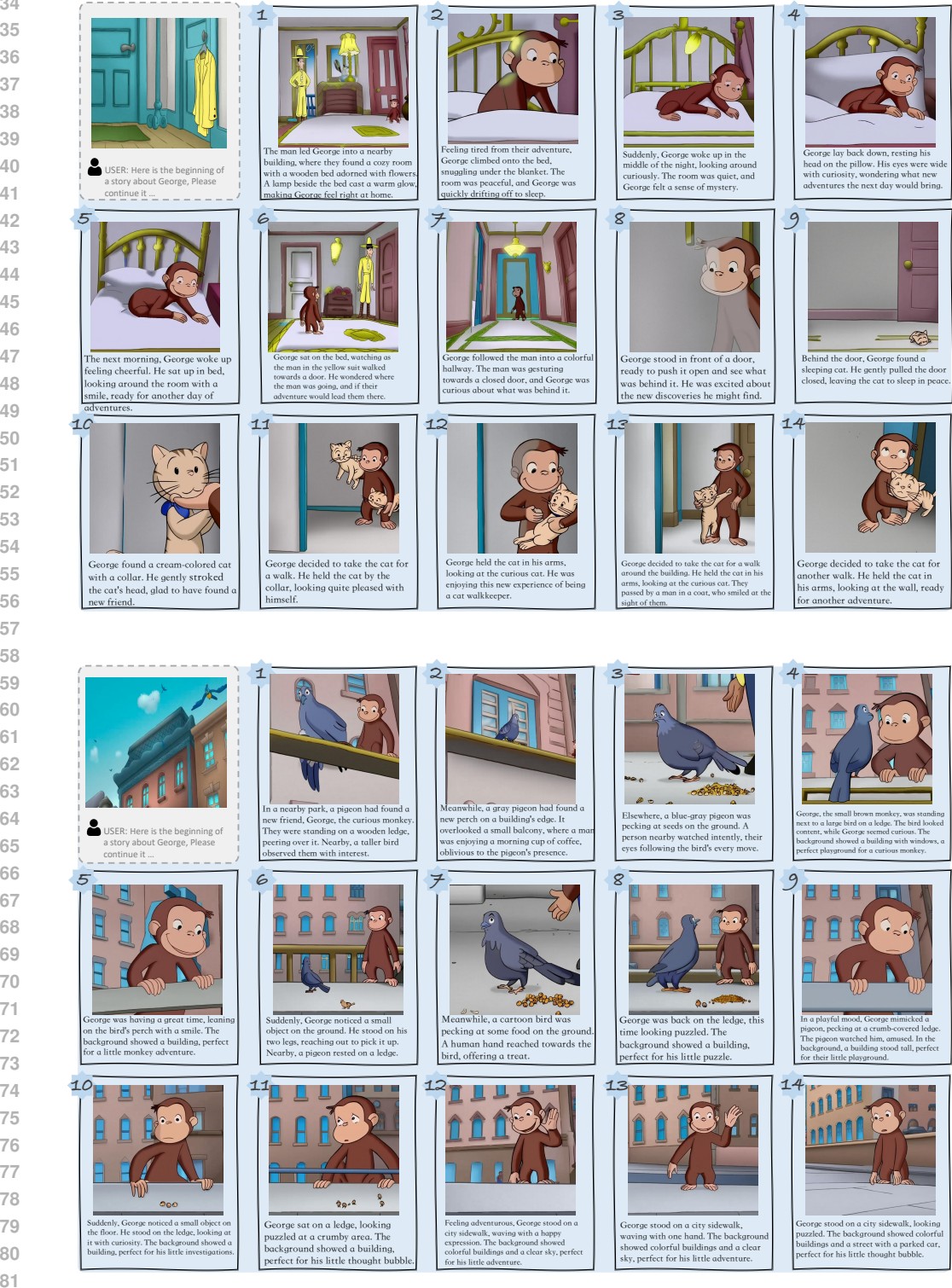

Figure H: Multimodal story generation results of SEED-Story.

story images / the engagement of the story / the coherence of the generated text and images}. Avoid any position biases and ensure that the order in which the responses were presented does not influence your decision. Do not allow the length of the responses to influence your evaluation. Do not favor certain names of

the assistants.Be as objective as possible. After providing your explanation, output your final verdict by strictly following this format: "[[A]]" if assistant A is better, "[[B]]" if assistant B is better, and "[[C]]" for a tie."

## G.2 SCORE EVALUATION

We also provide a prompt for directly estimating the performance of the generated results without comparing to others. The prompt we used is shown below. We present the direct estimation score is shown in Table 7

"Please act as an impartial judge and evaluate the quality of the generation story contents provided by an AI assistant. Your job is to give a score out of 10. Your evaluation should consider **{the style consistency of the story images / the engagement of the story / the coherence of the generated text and images}**. Do not allow the length of the responses to influence your evaluation. Be as objective as possible. After providing your explanation, output your final score by strictly following this format: "[[score]]", such as "[[7]]"."

Table 7: GPT4 score evaluation results in 3 different aspects-style consistency, story engaging level, and text-image coherence.

|  | Style ↑ | Engaging↑ | Coherence↑ |
|---|---|---|---|
| **SEED-Story** | 8.61 | 6.27 | 8.24 |

## H  STORY VIDEO

To showcase the capabilities of our multimodal generation model, we employ a video generation technique to animate the images. We then synchronize these moving images with audio to create a narrative video, which is available in our supplementary materials.

## I  DATA USAGE AND LICENSE

### I.1  CURIOUS GEORGE

Curious George is an animated series featuring George, a curious monkey whose adventures teach preschoolers about math, science, and engineering. Guided by The Man with the Yellow Hat, George explores the world through problem-solving and experimentation, making it a delightful and educational experience for young viewers.

Curious George is released on PBS KIDS PBS Kids (2024a;b), a not-for-profit institution. It is a production of Imagine, WGBH and Universal. Curious George and related characters, created by Margret and H.A. Rey, are copyrighted and trademarked by Houghton Mifflin Harcourt and used under license. Licensed by Universal Studios Licensing LLC. Television Series: ©2024 Universal Studios. The terms of use of them are provided in `https://www.pbs.org/about/about-pbs/terms-of-use/`.

Our usage fully comply with the terms of use. 1) Personal Uses Permitted: My project is non-commercial and educational, which aligns with personal uses as outlined by PBS. we are not using the information for commercial purposes or exploiting it in a manner inconsistent with PBS rules. The use is strictly for educational and research purposes within an academic setting. 2) User's Obligation to Abide By Applicable Law: We will ensure all research activities comply with local laws, particularly those relating to copyright and intellectual property rights. Our use will not involve unauthorized reproduction, distribution, or exhibition that violates Intellectual Property Laws. All data are for research only. 3) Content of Information: We will responsibly use the "Curious George" materials, ensuring that all content used in our research is accurately cited and acknowledged. Any PBS content incorporated into your project will be clearly attributed to PBS.

### I.2 RABBIDS INVASION

"Rabbids Invasion" is a French-American computer-animated TV series that breathes life into the zany antics of Ubisoft's popular Rabbids video game characters. Created by Jean-Louis Momus and featuring the voice of Damien Laquet, the show is a dynamic blend of humor and adventure tailored for a family audience. Since its debut on August 3, 2013, on France 3, the series has enjoyed multiple seasons and a global reach. The Rabbids are mischievous rabbit-like creatures whose escapades lead them into all sorts of unpredictable and hilarious situations, making "Rabbids Invasion" a delight for both kids and adults alike. Thanks to their release, we derive some subsets from the cartoon series Rabbids Invasion Animaj (2024b;a).

### I.3 THE LAND BEFORE TIME

The Land Before Time, an iconic animated film series created by Judy Freudberg and Tony Geiss and distributed by Universal Pictures, debuted in 1988 with significant contributions from Don Bluth, George Lucas, and Steven Spielberg. This franchise, consisting of an initial film followed by 13 sequels, a TV series, video games, and extensive merchandising, explores the adventures of five young dinosaurs who learn key life lessons about friendship and teamwork through their prehistoric trials. Despite the absence of the original creators in the sequels, the series has continued to captivate audiences, emphasizing themes of community and perseverance across its extensive narrative arc. Thanks to their release, we derive some subsets from their websites TheLandBeforeTime (2024a;b).

### I.4 APPRECIATION

Leveraging the data derived from "Curious George," "Rabbids Invasion," and "The Land Before Time," we have significantly advanced the capabilities of our story generation models. This progress has direct and impactful implications for children's education by enhancing their imaginative faculties and fostering a keen interest in learning. By integrating elements from these animated series into our models, we not only engage young minds but also deepen their affection for animated storytelling. Consequently, this not only meets but also amplifies educational objectives, such as improving literacy and cognitive skills through enjoyable and interactive content. The successful application of data from these beloved animations in our research exemplifies how academic pursuits can harmoniously blend with educational entertainment, ultimately delivering multifaceted benefits that extend well beyond conventional learning environments.

Lastly, we extend our profound appreciation to the creators and maintainers of "Curious George," "Rabbids Invasion," and "The Land Before Time," each a rich and vibrant resource that has significantly contributed to the scope and success of our research. The engaging narratives and characters from these series, especially the ever-curious George, the mischievous Rabbids, and the adventurous dinosaurs from The Land Before Time, have provided invaluable data that enhanced our narrative generation models. This project benefited immensely from the educational and entertaining content crafted with meticulous attention to detail, fostering imagination and learning in young audiences. We acknowledge the pivotal role that these animated series have played in advancing academic research aimed at educational technology. The commitment of the teams behind these beloved series to fostering curiosity and learning is both inspiring and exemplary. We are immensely grateful for the opportunity to incorporate such cherished resources into our scholarly work.

## J  BROADER IMPACTS

This project may potentially produce copyrighted content, particularly when used inappropriately or without adherence to existing intellectual property laws. To mitigate this risk, we will implement a rigorous compliance framework that respects the copyrights of third parties. This involves setting strict usage licenses that align with the legal standards dictated by our data sources. Our aim is to protect intellectual property rights while fostering innovation and ethical use of our technology. We also commit to educating users on the importance of respecting intellectual property rights when using our technology. This will be achieved through detailed user guidelines, training sessions, and readily available support to help users understand and navigate the complexities of copyright laws. By taking these measures, we aim not only to comply with legal standards but also to promote a

culture of respect for intellectual property within our user community, thereby contributing positively to the broader digital ecosystem.

## K    LIMITATIONS

**Lack of Realistic Data Experimentation:**    This limitation points to a potential gap in the validation of the SEED-Story model under practical, real-world conditions. Without experiments using realistic data, it's difficult to ascertain how the model would perform in scenarios that are not perfectly controlled or that deviate from the training conditions. This can be crucial, especially in applications like storytelling where the context and variability of real-world data play significant roles. A possible solution would be to incorporate a broader range of test conditions, including noisy data or data from "in-the-wild" storytelling scenarios, to evaluate the robustness and adaptability of the model.

**Training on a Non-Diverse Dataset:**    The second limitation is the restriction of the model's training to animation datasets which does not cover a large scale or diverse styles. This can severely limit the model's ability to generalize and produce outputs in styles that are not represented in the training data. This is particularly limiting in creative tasks such as storytelling where the ability to adapt to various artistic and narrative styles is crucial. To mitigate this, expanding the dataset to include a wider array of styles, genres, and visual aesthetics could be beneficial.

