# OpenReview forum: "SEED-Story: Multimodal Long Story Generation with Large Language Model"
_ICLR.cc/2025/Conference — ICLR 2025 Conference Withdrawn Submission_

### Official Review · Reviewer_wEr9 · 2024-10-29

**Soundness:** 4
**Presentation:** 3
**Contribution:** 4
**Rating:** 6
**Confidence:** 5

**Summary:**

The paper presented SEED-Story, aiming to generate long, coherent stories combining narrative texts with visually consistent images, addressing challenges in multimodal story generation. It proposes a multimodal attention sink mechanism for efficient long-sequence generation, and release the StoryStream dataset to benchmark the model. Through comparison and user studies, the authors claim SEED-Story outperforms existing models in terms of coherence, visual quality, and style consistency in multimodal storytelling.

**Strengths:**

+ The paper curated a high-resolution long story dataset StoryStream that contains 257k images with story length up to 30 frames.
+ It proposes a multimodal attention sink mechanism for efficient long-sequence generation.
+ The qualitative results are impressive for generating long multimodal stories.

**Weaknesses:**

+ The model architecture is quite similar SEED, with proposed multimodal attention sink.
+ The evaluation is not very comprehensive, giving existing literature also evaluates image consistency and character accuracy in the story generation setting.

**Questions:**

+ Why attention sink always choose the beginning of the text/image tokens, what is the insight behind this. How about long sequence and the current generation is not quite relevant to the beginning.
+ The evaluation metric is not very comprehensive for the story generation setting. It only shows FID and CLIP score. First of all, CLIP score is not very reliable for identifying the fine-grained vision-language similarity. Except for the human evaluation, is there any metric for demonstrating the image consistency and character accuracy?
+ The paper targets on multimodal story generation setting. How to demonstrate the text generation follows a coherent and logical storytelling progression?

---

### Official Review · Reviewer_r98Z · 2024-11-03

**Soundness:** 3
**Presentation:** 3
**Contribution:** 3
**Rating:** 6
**Confidence:** 4

**Summary:**

This work tackles the challenging problem of multimodal story generation – generating coherent textual stories with corresponding images interleaved with them. The work proposes a three-stage procedure, where the first stage trains an image de-tokenizer to regress towards target image features, while the second stage unifies the visual tokenizer and detokenizer with a multimodal LLM to generate simultaneously the image and textual features. Then, the trained model will undergo a final de-tokenizer adaptation step where the diffusion loss will be used to refine the outputs in their pixel quality. The authors also extend the attention sink mechanism with “focused” tokens on several special positions to preserve story length and consistency.
The work features additionally a collected multimodal story generation dataset, which is also used to evaluate the above proposed method.

**Strengths:**

- The tokenizer and de-tokenizer training coupled with multimodal instruction finetuning is neat.
- The adaptation stage seems useful.
- The extended multimodal attention sink mechanism is interesting and should be studied more.

**Weaknesses:**

- While the proposed method is evaluated well on the featured StoryStream dataset, the method should still be evaluated on existing ones such as Pororo and Flint Stones.
- While I appreciate the proposed dataset, the visual consistency test is still lacking and a well defined metric particularly for visual consistency is much needed.
- The attention sink mechanism is not well-studied yet in this manuscript, for example, how does the performance with its introduction scale with lengths, story complexities, and character consistencies?
- Some human evaluation is needed for the results comparisons.

**Questions:**

- How do ensure the quality of the proposed dataset, and what are the criteria measured if any?

---

### Official Review · Reviewer_7aJt · 2024-11-04

**Soundness:** 2
**Presentation:** 4
**Contribution:** 2
**Rating:** 3
**Confidence:** 3

**Summary:**

The authors present a new method for story generation (seed-story), including the generation of images along with text. The model to generate the stories is based on a multimodal LLM. Images are tokenized with a ViT based tokenizer. Then images can be fed into an LLM to predict the next tokens, which could be images or text. A detokenizer, based on SDXL is then used to generate viewable images from the tokens. The image detokenizer and the LLM are finetuned for this specific task, with LORA being used for the LLM.

The authors introduce a new dataset, StoryStream, which consists of interleaved image/text stories to train their model. To create the dataset, the authors first sample keyframes and corresponding subtitles from children's cartoons. Then a caption model is used to caption frames. Finally GPT-4 is used to generate a consistent story given the subtitles and captions. The final dataset is larger in terms of story length and higher resolution compared with existing datasets.

Seed-story also attempts to generate longer stories. Long stories are difficult to generate given the limited training data. Therefore the authors use a "train-short-test-long" approach to generate longer videos. The authors introduce a variation on attention-sink for multimodal data. Attention sink uses a sliding attention window plus a few initial tokens at the beginning of the sequence. The proposed multimodal attention sink also includes the tokens corresponding to the beginning and end of the image tokens. It is shown that this approach generates much higher quality story images.

Seed-story is primarily compared with MM-interleaved, another approach that can be used to generate stories. Compared with MM-interleaved seed-story achieves lower FID, better style consistency, story engagement, and image-text coherence.

**Strengths:**

The strongest contribution of the paper is the new dataset (StoryStream). There seems to be a need for this new dataset, given that the existing datasets look quite low-resolution and simple. It will be useful for other papers to use this dataset to train and evaluate.

The major technical contribution of the paper is the multimodal attention sink. This is quite an interesting non-obvious observation that the model attributes high attention to tokens near BoI / EoI. Then the authors take this observation and propose a multimodal attention sink which seems to be useful for generating longer stories. This contribution could be useful in other problems outside of story generation.

**Weaknesses:**

The long generated stories don't seem to be very good in terms of storytelling. For example, Figure F does not really suggest any logical story (e.g. with a beginning, middle, end). It seems more like random captions were generated, especially near the end. The llm also starts repeating itself often (e.g. "A man in a hat and shirt looked surprised", "George, the cartoon monkey, stood in a grassy area". The authors in the paper say that this plot is "engaging", but IMO it is not engaging.

This is a difficult task, but a simple baseline is to have a text only LLM generate a curious george story. The story in Figure F looks quite poor compared to a pure text baseline. Another simple baseline would be to have a text-llm generate a story and then have a diffusion model generate an image for each paragraph. This might lack consistency between images, but overall might be better, especially if rated by humans. It feels like there is potential in this paper but I feel more iteration is needed to generate reasonable stories.

I'm also wondering how good the StoryStream dataset actually is for this task. From figure 5 it actually doesn't seem to have a very coherent story either, which might be why the trained model struggles. Figure 5 looks like a bunch of image captions with very little narrative storytelling. To be fair, it does seem that the other existing datasets have the same issue.

The main comparison with previous works is with MM-interleaved, but I am concerned about the fairness of these comparisons. MM-interleaved uses stable-diffusion-2-1 and Seed-story use SDXL. SDXL should be much better than stable-diffusion-2-1, and it is possible that the results in Figure 6a,b,d are strongly affected by this. Also, MM-interleaved use ViT-L and seed-story uses ViT-G as the image encoder which might also affect these numbers. MM-interleaved uses Vicuna-13b and seed-story uses llama2 which could affect Figure 6c. The authors do not mention these important differences between MM-interleaved and seed-story in the paper.

Nitpicks:
References need to be cleaned up, there are multiple cases of duplicate references. E.g. for Qwen-vl, "Improved baselines with visual instruction
tuning"

**Questions:**

1. For Figure 8, at what size sequence length does the sliding window approach stop working? I would also like to know for Table 2 what sequence length this is?

It would be interesting to show the FID/clip score at different sequence lengths. E.g. for seq length = 10 I expect all the methods are equivalent, but I am not sure how the numbers compare with seq length = 20, 30, 40, 50 . Does the multimodal attention sink only help when the test seq length is much larger than the training?

2. For section 5.1 please clarify the differences between MM-interleaved and seed-story (e.g. architecture/base models)

3. Does the mm attention sink affect the quality of the text? Figure 8 / Table 2 are mostly concerned with the quality of the images

---

### Official Review · Reviewer_XsKC · 2024-11-07

**Soundness:** 2
**Presentation:** 3
**Contribution:** 2
**Rating:** 3
**Confidence:** 3

**Summary:**

This paper tackles the task of multimodal story generation which generates both story text and corresponding images from give input image and text, by proposing SEED-Story model utilizing Multimodal Large Language Model. Additionally, authors extend attention sink mechanism and introduce a multimodal attention sink mechanism to enable the model to generate long sequences in an efficient manner. Finally, a new dataset called StoryStream was proposed for training and benchmarking the task of multimodal story generation with longer story length. In experiments, it has been shown that the proposed approach outperforms the baseline, MM-interleaved (Tian et al. (2024)), in both quantitative and qualitative evaluations.

**Strengths:**

- The proposed dataset has high-quality and high-resolution images with longer story length, which would be useful for research community.
- The proposed model seems to make sense, and the presented analysis for multimodal attention sink mechanism is nice.
- The experimental results show that the proposed approach outperforms the baseline on the proposed dataset and demonstrate its effectiveness even though it still has a limitation (please see the weaknesses below).

**Weaknesses:**

- The novelty of proposed approach seems somewhat incremental as it looks to share some ideas with MM-interleaved (Tian et al. (2024)). Also the proposed multimodal attention sink mechanism seems just a slight modification of existing attention sink mechanism even though it includes nice analysis.
- The proposed approach was only evaluated on the proposed StoryStream dataset. However, as MM-interleaved (Tian et al. (2024)) was also evaluated on both Pororo and Flintstones datasets in their paper, it would make the paper stronger with additional evaluations on more datasets.
- Another major concern is baseline choice. The authors just chose MM-interleaved, but I think it is still possible to use more natural baselines. e.g., generating story text first and then generate corresponding images with existing story visualization approaches.
- The experimental result is not so convincing. e.g., in Fig. 7 (bottom), in the generated image sequence, the hat appeared and disappeared, and the color of hat was changed, so its consistency seems still not very good.

**Questions:**

- How may participants were involved in user study presented in Section A of Appendix?
- My understanding is dense attention uses attentions between all tokens, and the proposed multimodal attention sink mechanism uses a subset of the tokens. So I think dense attention may perform (almost) equally well as multimodal attention sink mechanism even though it is more expensive. Can you clarify this?

---

### Official Review · Reviewer_r6Eh · 2024-11-09

**Soundness:** 2
**Presentation:** 2
**Contribution:** 3
**Rating:** 3
**Confidence:** 4

**Summary:**

The authors propose a method, SEED-Story, for the task of image-text multimodal story generation.
They design architectures that tokenize images for tuning large language models (LLMs) and detokenize them for image generation. They also employ a multimodal attention sink to facilitate long story generation.
They evaluate experimentally the quality of generated images and assess story quality with a baseline method, MM-interleaved.
Additionally, a novel dataset, StoryStream, is introduced.

**Strengths:**

1. The samples of generated outputs demonstrated in the manuscript look impressive, especially in long story generation, character consistency, and narrative text quality.
    This work would be significant when these examples objectively represent general performance.

2. The introduction of the StoryStream dataset is a valuable contribution that could advance future research in this field.
    The authors present StoryStream Dataset, which is already downloadable at https://huggingface.co/datasets/TencentARC/StoryStream.

**Weaknesses:**

1. This paper lacks essential elements such as a clear motivation and problem definition, making it difficult to judge whether the research goal is successfully achieved.
    Also, those make readers confused about whether the evaluation metrics and comparative methods are appropriate or sufficient.

2. In order to validate academic value, the manuscript should provide evidence theoretically or experimentally.
   The manuscript shows the quality of generated images with FID and CLIP score as an ablation study, and relative comparison story quality with a baseline method, MM-interleaved.
    I think the materials above are NOT enough to clarify the validity of improvement to keep fair comparison and reproducibility.

The results shown are promising, but clearer motivation, explicit research questions, and rigorous comparative experiments would strengthen the paper. The contribution of a new dataset is notable, yet the lack of comprehensive evaluation limits the ability to objectively assess the work's significance.

**Questions:**

See the weaknesses

**Details Of Ethics Concerns:**

At the review stage, I could easily find the corresponding materials of this submission : (1) GitHub repository is publicly available at https://github.com/TencentARC/SEED-Story, and (2) arxiv paper at https://arxiv.org/ab/s2407.08683.

Even it seems rather popular these days,
user double-blind review, I think that there is a possibility of making effects on reviewers biased in any forms.

---

### Note · Authors · 2024-11-13

I have read and agree with the venue's withdrawal policy on behalf of myself and my co-authors.